# Role of Aquaporins in the Physiological Functions of Mesenchymal Stem Cells

**DOI:** 10.3390/cells9122678

**Published:** 2020-12-13

**Authors:** Antonella Zannetti, Gheorghe Benga, Arturo Brunetti, Francesco Napolitano, Luigi Avallone, Alessandra Pelagalli

**Affiliations:** 1Institute of Biostructure and Bioimaging, CNR, Via T. De Amicis 95, 80145 Naples, Italy; antonella.zannetti@ibb.cnr.it; 2Romanian Academy, Cluj-Napoca Branch, Strada Republicii 9, 400015 Cluj-Napoca, Romania; gbgbenga@gmail.com; 3Department of Advanced Biomedical Sciences, University of Naples Federico II, via Pansini 5, 80131 Naples, Italy; arturo.brunetti@unina.it; 4Department of Veterinary Medicine and Animal Production, University of Naples Federico II, via Veterinaria 1, 80137 Naples, Italy; francesco.napolitano3@unina.it (F.N.); luigi.avallone@unina.it (L.A.); 5CEINGE-Biotecnologie Avanzate, Via Gaetano Salvatore 486, 80145 Naples, Italy

**Keywords:** aquaporins, water channel, mesenchymal stem cells, physiology, cell migration, cell differentiation

## Abstract

Aquaporins (AQPs) are a family of membrane water channel proteins that control osmotically-driven water transport across cell membranes. Recent studies have focused on the assessment of fluid flux regulation in relation to the biological processes that maintain mesenchymal stem cell (MSC) physiology. In particular, AQPs seem to regulate MSC proliferation through rapid regulation of the cell volume. Furthermore, several reports have shown that AQPs play a crucial role in modulating MSC attachment to the extracellular matrix, their spread, and migration. Shedding light on how AQPs are able to regulate MSC physiological functions can increase our knowledge of their biological behaviours and improve their application in regenerative and reparative medicine.

## 1. Introduction

Aquaporins (AQPs) are a family of transmembrane proteins that form water channels and work as regulators of intra- and inter-cellular water transport [1]. To date, thirteen AQPs that are widely distributed in specific cell types of various tissues have been characterised [2]. The major roles of AQPs have been investigated in both physiological and pathological conditions, and the results highlight their involvement in the transfer of water, gases, and small solutes (urea and glycerol), to maintain cell homeostasis [3,4,5]. These proteins regulate many biological processes through their intrinsic activity including cell proliferation, migration, apoptosis, and mitochondrial metabolism.

In addition, several studies have focused on the involvement of AQPs in intriguing aspects of cell biology and have demonstrated that they are involved in a variety of physiological processes and pathophysiological conditions [4,6,7,8].

Here, we review the current understanding about the roles played by AQPs in mesenchymal stem cell (MSC) functions and highlight their involvement in stem cell proliferation, migration, and differentiation.

Specific features of MSCs rely on their self-renewal ability, low immunogenicity, and the ability to migrate, proliferate, and differentiate in different cell types [9,10]. Notably, the biological activities associated with MSCs migration and proliferation are of particular importance because they are involved in tissue regeneration. Following tissue damage, MSCs are able to mobilise from the tissue of origin and migrate through the peripheral circulation to the injured site, where they proliferate and differentiate, thus facilitating the healing process through the activation of various mechanisms [11].

Such processes require the orchestration of multiple signals induced by mechanical (hemodynamic forces applied to the vessel walls through shear stress, vascular cyclic stretching, and extracellular matrix stiffness) and chemical factors (chemokines, and growth factors), that can act simultaneously. MSCs can migrate through 3D tissue and regulate forces that induce cell deformation and act on physical tissue constraints from the mechano-microenvironment [12].

It has also been demonstrated that the tissue source, growth factors, ageing, the microenvironment, and hormones can influence the MSC proliferation rate. In particular, in vitro and in vivo studies have shown that the tissue source and aging affect the properties of MSCs, including their proliferative capacity, lifespan, and ability to differentiate efficiently [13,14,15]. In addition, Zhu et al. observed that a hypoxic microenvironment can increase the proliferation of placenta-derived MSCs via the mitogen-activated protein kinase (MAPK)/extracellular signal-regulated kinase (ERK) pathway [16]. These aspects, along with the various growth factors, cytokines, exosomes, and microvesicles secreted by MSCs [17,18] should be taken into consideration when designing strategies to enable the efficient use of MSCs for repairing dysfunctional organs.

## 2. AQPs

AQPs are a family of thirteen integral-membrane water channel proteins (AQP0 to AQP12) found in humans, animals, and plants. They can be classified into three main functional subfamilies based on their ability to facilitate transport: AQPs, aquaglyceroporins, and a third family that is comprised of AQPs with uncharacterised functions [2,19] (Table 1).

AQP0, AQP1, AQP2, AQP4, AQP5, AQP6, and AQP8 are considered as the classical water-specific channel proteins. In contrast, AQP3, AQP7, AQP9, and AQP10, which belong to the aquaglyceroporins sub-family, are characterised by their permeability to water, glycerol, urea, and a few small neutral solutes [20]. AQP11 and AQP12 represent the most distantly related paralogs as they show low amino acid sequence identity with the other AQP family members [21]. These AQPs show non-canonical subcellular localisation and functions.

To date, two functional motifs, namely, Asn-Pro-Ala (NPA) [22,23,24,25]) and the aromatic-arginine regions have been identified in the AQP sequence. The NPA region, located in the middle of the channel, is involved in proton exclusion and contributes to the localisation of AQPs to the plasma membrane (Figure 1). The aromatic-arginine region acts as a selectivity filter on the extracellular side of the AQP channel and blocks the entry of molecules larger than water.

AQPs are structurally organised as homotetramers, where each monomer (ranging from 26 to 34 kDa) is primarily composed of six transmembrane domains such that both the amino- and carboxy-termini of the protein lie inside the cell. The highly conserved NPA motifs are located in cytoplasmic loop B and in extracellular loop E [26]. These NPA motifs contribute to a monomeric pore structure that facilitates selective, bi-directional, and single-file transport of water in the classical AQPs [23] and water and glycerol in aquaglyceroporins [27].

AQPs are mainly localised to the plasma membrane, and to a lesser extent in cytosolic compartments, where they can be transported to the plasma membrane in response to hormones and kinase activation [28]. This localisation allows AQPs to regulate distinct processes occurring in different cellular compartments. Studies performed in humans and other animal species have shown that AQPs participate in a wide range of physiological functions, including water/salt homeostasis, exocrine fluid secretion, and epidermal hydration. Moreover, AQPs contribute to pathological conditions such as glaucoma, cancer, epilepsy, obesity in which water and small solute transport may be involved [1,24]. Of note, AQPs seem to act as modulators of adipocyte biology in obesity by facilitating glycerol release from adipose tissue. Mechanisms associated with cell regulation as well as cell migration and signalling have been linked to AQPs in cancer biology and metastasis development.

Molecular evidence has highlighted the importance of AQPs in facilitating multiple cellular processes such as (1) transepithelial fluid flow regulated by osmotic water transport across cell membranes, (2) cell migration and neuroexcitation, (3) cell proliferation mediated by glycerol transport, (4) adipocyte metabolism, and (5) epidermal water retention [1,6,29].

In response to environmental stimuli, AQPs regulate the flow of water and small molecules including glycerol, ammonia and urea in a tissue-specific manner to maintain their homeostasis (Table 2).

Water and fluid transport across the cell membrane are an important prerequisite for maintaining cell homeostasis and volume, which are factors that regulate several cell functions. Hoffmann and co-workers showed the importance of AQPs in the biological processes associated with changes in cell volume such as migration, inflammation, proliferation and cell death [51].

In particular, it has been observed that several factors (osmolality, modifications in the transmembrane ion gradients that generate an osmotic imbalance) that regulate cell volume through AQPs also influence processes that affect cell proliferation [52,53,54,55,56].

Of note, many findings support the hypothesis that AQP expression correlates with the different phases of the cell cycle [57,58,59,60,61]. The overexpression of AQP1 and AQP3 has been shown to affect the expression of essential checkpoint proteins, such as cyclins, and modify the levels of transcription factors and cytokines [62,63]. Delporte et al. [57] showed the involvement of AQP1 in the cell cycle regulation of an epithelial cell line. High levels of AQP1 mRNA and protein expression were found during the G_0_/G_1_ phase, whereas a significant decrease in mRNA and protein expression was observed in the S and G_2_/M phases. Moreover, AQP2 expression has been shown to accelerate the proliferation and cell cycle progression of the renal cortical collecting-duct cells by decreasing the transit time through the S and G_2_/M phases, possibly by increasing the cell volume [60].

Additionally, the altered function of AQPs is often correlated with pathological conditions. Nephrogenic diabetes insipidus and congenital cataracts are genetic diseases caused by loss-of-function mutations in AQPs [64,65]. Pathogenic autoantibodies against astrocyte specific AQP4 contribute to the development of the neuroinflammatory demyelinating disease, neuromyelitis optica [29,66]. Recent studies have shown an association between AQP polymorphisms and diseases such as cognition-related disorders, Alzheimer’s [67], and mesothelioma [68]. While the functional link between AQPs and these diseases needs to be investigated further, these data suggest that AQPs can be used as potential biomarkers for severe and disabling diseases. Moreover, there is a great interest in the development of therapeutic strategies using small-molecule modulators with the ability to target AQP function.

## 3. MSCs

MSCs are an exclusive class of plastic-adherent, adult stem cells present in most body tissues that are characterised by their ability to self-renew and differentiate into a variety of mesenchymal tissues, including bone, cartilage, adipose tissue, neurons, and haematopoietic cells [69,70].

The primary sources of human MSCs include the bone marrow (BM) [71], adipose tissue [72,73], umbilical cord [74] and placenta [75] (Figure 2).

In the last decade, MSC populations have been discovered in other tissues like skeletal muscle [76], dermal tissue [77], olfactory bulb [78], and dental pulp [79]. Recently, they have also been found in tumour microenvironments, thus supporting tumour progression [80,81,82,83] through the stimulation of mitogen and stress signalling as well as potentiating resistance to treatments [84,85,86].

MSCs normally show a high rate of cell proliferation, which makes them extremely useful in regenerative medicine and tissue engineering. Furthermore, the use of MSCs is associated with low tumorigenicity and relatively few ethical restrictions. MSCs show pleiotropic activity on different body tissues and can be easily isolated from different autologous or allogeneic sources [87]. These properties make MSCs a potential and promising candidate in the field of regenerative medicine.

While the application of MSCs in musculoskeletal disorders [88,89] and other degenerative diseases [90] is well-studied, the other more complex approaches require further analysis of the MSC physiology. Furthermore, analysis of the abilities of MSCs in regenerative medicine using in vivo approaches has shown that tissue regrowth is not exclusively linked to MSC (trans)differentiation, but rather to autocrine and paracrine signalling factors secreted by MSCs in response to local stimuli [91], growth factors [92] and inflammatory mediators [93]. Future studies aimed at designing functional tissues or organs should focus on cell proliferation and the related regulatory mechanisms, and on the identification of cell proliferation markers rather than cell cycle activity.

Recently, it was reported that intracellular potassium (K^+^) is involved in human MSC proliferation and cell cycle regulation. Marakhova et al. [94] showed that a decline in K^+^ levels is associated with an accumulation of cells in the G_1_ phase and a delay in proliferation. Moreover, K^+^ levels have also been found to be correlated with MSC age, which highlights the importance of this ion in stem cell proliferation and its potential application as a biomarker.

Notably, K^+^ has been shown to act as a key player with an active role in cell cycle progression and cell volume adjustment [53,95,96,97,98]. Using the K^+^ channel blocker tetraethylammonium (TEA), it was found that a decrease in the inward Rb+(K+) leakage and a delay in the cell cycle were associated with a decrease in cellular K^+^ content per milligram protein. These results suggest that cycling MSCs have a higher water content per milligram protein than quiescent or differentiated cells.

However, it is important to bear in mind that the proliferative ability of MSCs depends on several variables, including tissue donor age, passage number, and plating density [99,100,101]. Additionally, the composition of the growth medium, including the presence of specific cytokines and growth factors, can stimulate MSC proliferation and affect the physiological properties of these cells such as motility, morphogenesis, and survival.

MSCs can differentiate into several cell types, including osteoblastic, chondrogenic, and adipogenic lineages. Recent studies showed that MSCs can also differentiate into neuronal and cardiomyogenic lineages. MSC differentiation is determined by a complex, multi-step process that is coordinated by specific regulators [102]. For example, Runx2 and Osterix are master regulators for osteogenic differentiation [103], while peroxisome proliferator-activated receptor γ (PPARγ) and CCAAT/enhancer binding protein β are important factors promoting adipogenesis [104]. Moreover, specific stimuli including bone morphogenetic proteins and wingless proteins (Wnts), magnetic field stimulation, and dexamethasone and ascorbate supplementation of osteogenic media can facilitate MSC differentiation into osteoblasts [105,106,107,108,109,110,111]. This selective ability to promote osteogenic differentiation has enormous clinical implications, thus supporting MSC application in the field of bone regenerative medicine, tissue engineering, and preventive therapies for bone metabolism diseases, such as osteoporosis and fracture healing.

MSCs can also be stimulated to differentiate into neurons expressing markers of nervous system cells, such as nestin, β-III tubulin, microtubule associated protein 2 (MAP2), and neuron-specific enolase 2 (ENO2) [112,113]. Using mouse adipose-derived MSCs, Pavlova et al. have shown that brain-derived neurotrophic factor (BDNF) or retinoic acid is more efficient in increasing neuronal marker expression compared to other routinely used protocols [114]; this highlights the importance of these factors in promoting cell differentiation. Again, further in vivo studies have shown that induced MSCs with BDNF or retinoic acid and transplanted into mouse brains have a higher migration rate compared to controls [112].

In response to treatment with specific factors, MSCs can also differentiate into endodermal lineages, including cardiomyocytes, hepatocytes, and ectodermal neuronal-lineage cells [73] through the mesengenesis mechanism. By means of this process, MSCs also give rise to myoblasts, bone marrow (BM) stromal cells, fibroblasts, and cells that comprise the connective tissue of the body including ligaments and tendons [115].

Cell migration is a complex process, orchestrated by signal transduction, cytoskeleton rearrangement, and morphogenesis and occurs through four distinct steps: cell polarisation, protrusion, adhesion and rear retraction [116]. Multiple mechanical and chemical factors act as regulators of cell migration, conveying signals from the extracellular matrix to the cellular adhesion complex, where the focal adhesion kinase (FAK) plays a crucial role [117]. Previous studies have shown that cell migration plays a pivotal role in regenerative medicine in guiding MSCs to the damaged sites and promoting regeneration. However, the use of chemo-attractants, like chemokines secreted by injury sites can further improve MSC recruitment to injured tissues, thus enhancing the natural healing process [118].

## 4. Physiological Roles of AQPs in Driving MSC Function

Research performed over the last decade has found that AQPs are expressed in several stem cell types, which suggests their involvement in a variety of physiological processes [119,120]. However, their exact functional role in MSCs remains to be completely clarified, albeit some aspects of AQPs role in regulating MSC behaviour in certain disorders (acute lung injury, hepatocarcinoma) have been studied using animal models of disease. These studies have facilitated a better understanding of the molecular pathways underlying MSC involvement in these diseases [121,122,123].

The presence of AQPs in the apical membranes of different MSCs, their co-localisation with other systems such as the Na^+^/K^+^ ATP pump, and the effect of AQP inhibitors including HgCl2 and TEA suggest that AQPs may regulate the migration and differentiation of MSCs [119]. The involvement of AQPs in various aspects of stem cell biology is related to their capacity to regulate the flux of fluids between the outer and inner cellular compartments to maintain homeostasis. Interestingly, the control of stem cell volume is an important prerequisite for the regulation of various stem cell properties, including proliferation, migration and differentiation.

Experiments using human BM-MSCs demonstrated that the attachment, spread, and migration of stem cells are accompanied by water efflux and cell volume reduction, which in turn, correlate with the strength of the attachment [124]. The introduction of the AQP1 gene increases the migration ability of MSCs. These results have been confirmed by in vivo experiments showing that the injection of MSCs expressing AQP1 in a rat tibia fracture model enhances bone healing compared to the injection of non-transfected MSCs. This study suggests that the AQP1-mediated improvement in MSC migration likely occurs through the modulation of β-catenin and FAK expression [125] (Table 3). In addition, our recent findings highlight the crucial role played by AQP1 along with C-X-C chemokine receptor type 4 (CXCR4) in regulating ovine MSC (oMSC) migration through the activation of the serine/threonine protein kinase (Akt) and extracellular signal-regulated kinase (Erk) intracellular signal pathways [126] (Table 3).

Accumulating evidence has shown that AQPs are involved in the function of the brain and central nervous system (CNS) by either maintaining water homeostasis or regulating cerebrospinal fluid secretion and absorption, as well as fluid transport across neutrophils, cell volume regulation, and central osmo-reception [31,138]. Among the AQPs, AQP1 [139], AQP4 [140,141,142], and AQP9 [143,144] play a central role in CNS-related diseases including cerebral oedema, brain tumours and epilepsy.

More recently, further studies have demonstrated that alteration in AQPs expression is the main cause of water balance dysregulation in the CNS at both the cellular and subcellular levels, and it is responsible for the pathogenetic profile of different diseases such as focal oedemas, brain tumours, brain ischaemia, and traumatic injuries [145,146,147,148]. The brain water content and swelling of the astrocyte foot have been found to be significantly reduced in a brain oedema mouse model that lacks AQP4 [149].

AQPs seem to play a specific role during neurogenesis as neural stem cells (NSCs) are able to move considerable amounts of water across the cell, thus rapidly changing the cell volume. La Porta et al. [133] (Table 3) showed that AQP8 (localised in mitochondria) regulates water balance in adult NSCs (ANSCs) by mediating the osmotic movement of water between the cytoplasm and the mitochondrial compartment. Interestingly, in vivo studies carried out on rat models have reported similar localisation of AQP8 in other cells of the brain, which suggests a possible role of this mitochondrial isoform in many physiological functions including metabolism and apoptosis, as well as in the pathogenesis of neurological diseases such as Parkinson’s [150]. Cavazzin et al. [130] used molecular and phenotypical characterisation to show that AQP4 and AQP9 expression is important in the differentiation of murine ANSCs present in the subventricular zone (SVZ). In particular, immunohistochemistry analysis demonstrated that ANSC-derived glial cells express low levels of AQP4 and display high levels of glial fibrillary acidic protein (GFAP), when astrocytes express high levels of AQP4 and low levels of GFAP. In contrast, while the occurrence of AQP9 is limited to a few cells, those cells that express AQP4 and AQP9 also co-express GFAP. These differences in the subcellular localisation of AQPs among the various ANSC-derived glial cells have led to the hypothesis that the physiological activity of AQPs is cell-specific (Table 3). Li et al. [151] showed that AQP4 is involved in NSC regulation and co-localises with gamma-aminobutyric acid A receptors (GABAARs) in the subependymal zone (SEZ). This result is intriguing considering that the activation of GABAARs induces hyperpolarisation and osmotic swelling in precursor cells, and thus promotes surface expression of the epidermal growth factor receptor and cell cycle entry. Other studies have investigated the role played by AQP4 in the proliferation, migration, and differentiation of ANSCs in vitro using AQP4-knockout mice [132]. Evaluation of connexin 43 and Cav.1.2 expression during the proliferation or differentiation of ANSCs showed that AQP4 in knockout mice causes significant downregulation of the expression of these proteins, which highlights the involvement of AQP4 in the self-renewal, migration and proliferation of ANSCs. In addition, the authors hypothesised that AQP4 regulates the migration of ANSCs through a multi-step process involving: (1) Ca^2+^ influx; (2) actin depolymerisation; (3) an increase in cytoplasmic osmolality; (4) cell membrane expansion with subsequent protrusion formation, and (5) actin re-polymerisation to stabilise the emerging protrusion. Taken together, these findings suggest that AQP4 can regulate the fundamental properties of ANSCs through the Ca^2+^-related signalling pathway [130] (Table 3).

The characterisation of the neural differentiation process of (Ad)-MSCs demonstrated the involvement of AQPs in this process and suggested their role in helping the cells to achieve rapid regulation of the volume during differentiation [128]. In particular, immunohistochemistry, Western blotting, and RT-PCR-based data showed that differentiated neuronal cells arising from sources other than Ad-MSCs, which express AQP1, also express AQP4, and AQP7 based on the cell type, thus suggesting a correlation between neural differentiation and the AQP expression profile during adult neurogenesis [128] (Table 3). Ma and co-workers [152] underlined the importance of AQP5 in modulation of the BM-MSCs differentiation using a mouse model of bone fracture. Accordingly, they found that a lack of AQP5 significantly increases the levels of adipogenic, osteogenic, and chondrogenic differentiation markers in mutant BM-MSCs [152]. Notably, BM-MSCs derived from knockout mice treated with the apoptotic drug, paclitaxel displayed an improvement in the bone healing process as well as a lower apoptosis rate, thus suggesting a modulatory role of AQP5 in slowing down BM-MSC differentiation.

In a previous study, it was reported that AQP1 and AQP3 expression is modulated during human MSC differentiation into chondrocytes [126]. These studies showed that during MSC differentiation, AQP1 and AQP3 expression increases substantially and is associated with high concentrations of collagen type II, aggrecan, and lubricin, thus demonstrating the relationship between these channel proteins and chondrocyte- ECM adhesion and migration [126] (Table 3).

Recently, Chen et al. [153] showed that AQP1 is expressed in rat tendon stem/progenitor cells (TSPCs) and that this expression is regulated during TSPC senescence through the JAK-STAT pathway. Similarly, Zhou J. et al. [120] found that AQP5 expression in human epidermal stem cells (EPSCs) decreases with skin ageing, suggesting that this channel protein has a critical role in regulating the balance between proliferation and differentiation [120] (Table 3).

AQPs are also regarded as key modulators of cancer stem cells (CSCs). Recent studies have shown that high levels of AQP expression in cancer cells and CSCs correlate with metastasis and resistance to therapies [154]. It has been reported that AQP3 affects the Wnt/GSK-3β/β-catenin pathway in liver CSCs (LCSCs), and modulates their stemness, differentiation, and apoptosis [135]. Other studies have identified AQP4 as the most abundant AQP in nasopharyngeal and lung CSCs and have shown its involvement in the regulation of large volumetric changes related to an increase in death rate [155]. Using a novel electro-osmotic microfluidic system that controls cell osmolarity gradients, AQP4 expression has been shown to be directly correlated with the speed of cell migration [137] (Table 3). In addition, when the expression of AQP4 is suppressed via siRNA, the CSC migration capability as well as the expression of stemness biomarkers such as Sox-2 or Oct-4 are significantly reduced. Other studies have shown that AQP0–AQP12 are highly expressed in human glioblastoma stem-like cell lines [130] and primary tumours [156,157]. Fossdal et al. [130] have described the involvement of AQP1, AQP4, and AQP9 in CSC function in glioblastoma and observed an upregulation of AQP9 levels and a downregulation of AQP1 and AQP4 levels (Table 3). These data indicate the specific role of AQP9 in facilitating CSC migration towards the surrounding normal tissues.

## 5. Conclusions

Recently, several studies have described the involvement of AQPs in regulating the functions of MSCs derived from different tissues. The classical role of AQPs in fluid transport has been postulated as a mechanism underlying the migration, proliferation, and differentiation of stem cells. Further studies elucidating how these water channels might participate in the control of stem cell functions in both physiological and pathological conditions could facilitate the use of MSCs in regenerative medicine.

## Figures and Tables

**Figure 1 cells-09-02678-f001:**
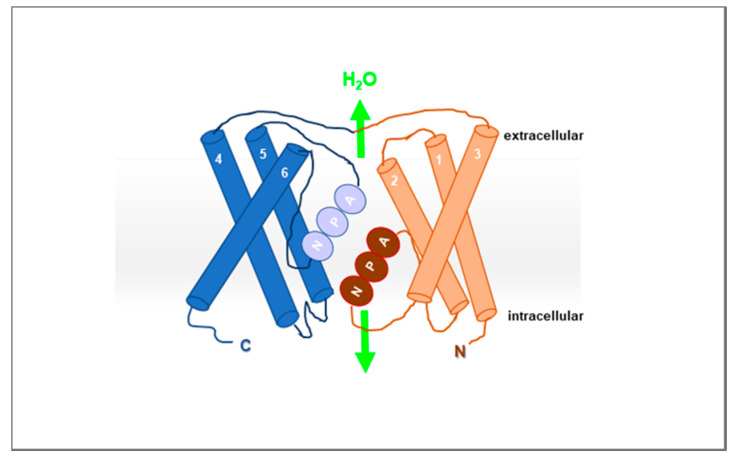
Schematic representation of the Asn-Pro-Ala (NPA) motif in the aquaporin (AQP) structure.

**Figure 2 cells-09-02678-f002:**
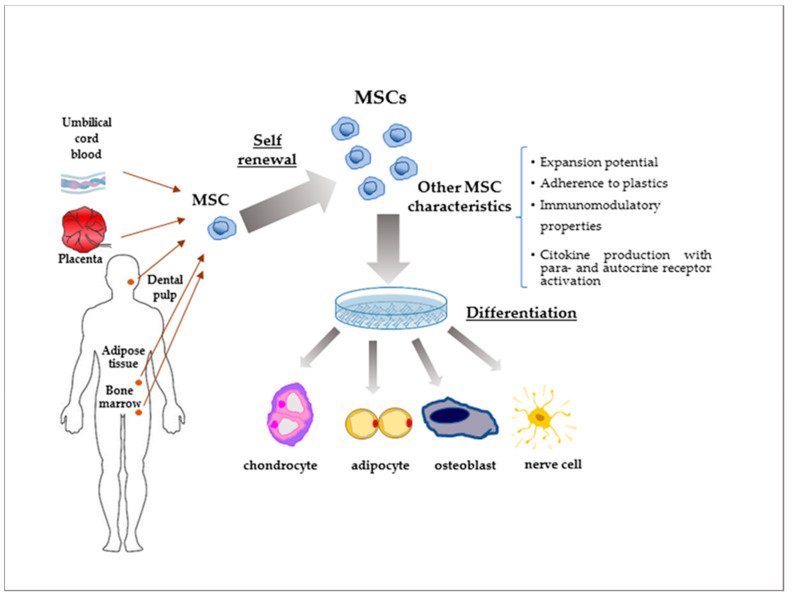
Main tissue origin of mesenchymal stem cell (MSCs) and their relative functions.

**Table 1 cells-09-02678-t001:** Classification and permeability characteristics of AQPs.

AQP Classification	Isoform	Permeability						
		H_2_O	Glycerol	NO	H_2_O_2_	NH_3_ and/or Ammonia	Urea	Uncertain
**AQPs**	AQP0	**+**	**/**	**/**	**/**	**+**	**/**	**/**
	AQP1	**+**	**/**	**+**	**+**	**+**	**/**	**/**
	AQP2	**+**	**/**	**/**	**/**	**/**	**/**	**/**
	AQP4	**+**	**/**	**+**	**/**	**/**	**/**	**/**
	AQP5	**+**	**/**	**/**	**+**	**/**	**/**	**/**
	AQP6	**+**	**+**	**/**	**/**	**+**	**+**	**/**
	AQP8	**+**	**+**	**/**	**+**	**+**	**+**	**/**
**Aquaglyceroporins**	AQP3	**+**	**+**	**/**	**+**	**+**	**+**	**/**
	AQP7	**+**	**+**	**/**	**/**	**+**	**+**	**/**
	AQP9	**+**	**+**	**/**	**+**	**+**	**+**	**/**
	AQP10	**+**	**+**	**/**	**/**	**+**	**+**	**/**
**Unorthodox-AQPs**	AQP11	**/**	**/**	**/**	**/**	**/**	**/**	**+**
	AQP12	**/**	**/**	**/**	**/**	**/**	**/**	**+**

Bold is refers to each subgroup of AQPs comprising a number of isoforms. The symbol + indicates that the specific AQP isoform is permeable to the specific molecule indicated.

**Table 2 cells-09-02678-t002:** Distribution and principal roles of AQPs in the main body tissues.

Tissue Localisation	Aqp Isoform	Main AQP Roles	Main References
**Brain**	AQP1, AQP3, AQP4, AQP6	- regulation of water homeostasis - control of osmotic pressure for efficient axonal conductance	[30,31]
**Eye**	AQP0, AQP1, AQP3, AQP4, AQP5,	- water balance maintenance in ocular tissues to ensure transparency in cornea and lens, in corneal wound healing - regulation of tear film osmolarity to produce aqueous humor, and to maintain retinal homeostasis	[32,33]
**Reproductive Tract**			
**Female**	AQP1, AQP3, AQP4, AQP5, AQP7, AQP8, AQP9, AQP10, AQP11	- secretive (vagina) and absorptive processes (utero) - maternal-fetal fluid exchange	[34,35]
**Male**	AQP1, AQP2, AQP3, AQP6, AQP7, AQP9, AQP11, AQP12	- fluid regulation for spermatogenesis, spermatozoa maturation and storage	[34,36,37]
**Heart**	AQP1, AQP3, AQP4, AQP6, AQP7, AQP9, AQP11	- contribution to the transcellular water flux across the endothelial membranes - involvement in the calcium signaling machinery at level of the cardiac and skeletal muscle	[38,39]
**Kidney**	AQP1, AQP2, AQP3, AQP4, AQP5, AQP6, AQP7, AQP11	- maintain normal urine concentration function, tissue development and substance metabolism	[40,41]
**Intestine**			
**Small**	AQP1, AQP3, AQP4, AQP5, AQP8, AQP9, AQP10	- involvement in fluid absorption and secretion	[4,42]
**Large**	AQP1, AQP3, AQP4, AQP5, AQP6, AQP7, AQP8, AQP9, AQP11	- regulation of water absorption	[4,42]
**Liver**	AQP1, AQP3, AQP7, AQP8, AQP9, AQP11	- canalicular and ductal bile formation	[43,44,45]
**Lung**	AQP1, AQP3, AQP4, AQP5	- regulation via transcellular pathway of water across the lung microvascular endothelium and epithelia	[46]
**Salivary glands**	AQP1, AQP3, AQP4, AQP5, AQP8	- saliva secretion process	[47,48]
**Skin**	AQP1, AQP3, AQP5, AQP9, AQP10	- hydration, wound healing, and skin epidermis homeostasis	[49]
**Stomach**	AQP1, AQP3, AQP4, AQP5, AQP7, AQP11	- contribution in secretion of gastric fluid	[42,50]

Bold indicates the tissue containing different Aqp isoforms.

**Table 3 cells-09-02678-t003:** Involvement of AQPs in different aspects of stem cell biology in health and diseases.

Stem Cell Type	Aqp Isoform	AQP Role	Reference
MSCs	AQP1	- Migration	[125,126,127]
	AQP1, AQP3, AQP4, AQP7	- Differentiation	[127,128]
progenitor MSC	AQP3	- Differentiation	[127,129]
glioblastoma stem-cell	AQP4, AQP9	- AQP9 involvement in the tumorigenicity process	[130]
NSCs	AQP4	- Proliferation, migration and differentiation	[131,132]
	AQP8	- Mitochondrial volume regulation during NSC differentiation	[133]
	AQP9	- Differentiation	[131]
EPSCs	AQP5	- Regulation of the balance between proliferation and differentiation	[120]
LCSCs	AQP3	- Proliferation and invasion - Stemness, differentiation and apoptosis	[134,135]
	AQP9	- Possible role in inducing CSCs death	[136]
Lung CSCs	AQP4	- Migration	[137]
CSCs	AQP4	- Migration	[137]

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
