# Peer review of "Role of Aquaporins in the Physiological Functions of Mesenchymal Stem Cells"

_cells, 2020, doi:10.3390/cells9122678_

Round 1
Reviewer 1 Report
The authors provide a useful overview of the role of aquaporins in stem cells.
However, several points need to be addressed.
- The text needs extensive language editing. Abbreviations such as MSC should be introduced. Words like “evidenced” are strange. Please revise.
- A scheme of AQP2 would be very helpful, as the authors refer to AQP segments such as the NPA motif. The manuscript would be more informative if such motifs would be visualized.
- Lines 44-56. It should be clarified that stem cells are not cancer cells. The was the text is written suggests they are. Please revise.
- Line 64 repeats line 33, and again line 103. There are repetitions throughout the manuscript. Please remove.
- In line 103, the authors write that AQP2 regulate flow of fluids. Please, be more correct and precise. What fluids are meant and how about ions?
- Line 80. Citation 19 is surely not the one describing the discovery of the NPA region. Please use appropriate reference.
- Line 126, please use a more suitable citation, e.g. by Bichet.
- Line 128, what is meant by “Other diseases,..”? Please say which ones are meant.
- A major criticism is that the manuscript is very superficial. For example, in line 179, what does “as reported by extensive literature” mean? Line 241, “This aquaporin seems to act modifying…? This sentence is basically without content because it leaves the reader without the actual result, such as the AQP2 regulates up or down or whatever is meant. The manuscript lacks precision throughout. Wordings like “high level” in line 241, “peculiar localization” in line 247, “dramatic” in line 248 or the sentence in lines 274-276, starting with “In particular,…”exemplify the point. Several times, “interestingly” is used. Science in general is interesting. Please delete. Such wording does not mean anything. The text requires extensive revision and introduction of not only qualitative but also of quantitative statements. Example, should also indicate the species.
- In the conclusion section, the authors link AQP2 and stem cells to regenerative medicine but have not addressed regenerative medicine in the manuscript. Therefore, the conclusion from line 281 is not justified.
Author Response
Comments by the reviewer#1
The authors provide a useful overview of the role of aquaporins in stem cells.
However, several points need to be addressed.
- The text needs extensive language editing. Abbreviations such as MSC should be introduced. Words like “evidenced” are strange. Please revise.
- According to the Reviewer #1 suggestions, the manuscript has been revised for English grammar by Elsevier English translate service (see Certificate attached). We modified some parts of the text by introducing abbreviations, as suggested by the reviewer. In addition, we changed the word “evidenced” with more appropriate words.
- A scheme of AQP2 would be very helpful, as the authors refer to AQP segments such as the NPA motif. The manuscript would be more informative if such motifs would be visualized.
- We appreciated the comment of the Reviewer #1 and added a chart illustrating AQP NPA motif (Figure 1). As suggested by the Reviewer, we think that this figure gives more detailed information to the reader.
- Lines 44-56. It should be clarified that stem cells are not cancer cells. The was the text is written suggests they are. Please revise.
- We thank the referee for her/his suggestion. In the revised version of the text we just referred to MSCs (now from line 43), rather than cancer stem cells.
- Line 64 repeats line 33, and again line 103. There are repetitions throughout the manuscript. Please remove.
- We thank the Reviewer for her/his comment and as suggested. Thus, whenever occurred, we managed to remove all the repetitions.
- In line 103, the authors write that AQP2 regulate flow of fluids. Please, be more correct and precise. What fluids are meant and how about ions?
- We totally agree with Reviewer’s comment. In the present version of the manuscript, we modified the line 103 (now line 133-134), specifying “flow of water and small molecules including glycerol, ammonia and urea”.
- Line 80. Citation 19 is surely not the one describing the discovery of the NPA region. Please use appropriate reference.
- We perfectly agree with the Reviewer’s comment, about the lack of more appropriate references for “NPA motif”. In this respect, we added a few ones (references 22-25, line 101).
- Line 126, please use a more suitable citation, e.g. by Bichet.
- We thank the Reviewer for her/his suggestion, and we cited line 126 (now 165) Bichet “D.G. Genetics and diagnosis of central diabetes insipidus. Ann. Endocrinol. 2012, 73, 117-127” (reference 64), in the new version of the manuscript.
- Line 128, what is meant by “Other diseases,..”? Please say which ones are meant.
- We thank the Reviewer for her/his comment. Here, we listed some of the main pathologies (cognition-related disorders, Alzheimer’s, and mesothelioma) associated to AQP polymorphisms (line 168).
- A major criticism is that the manuscript is very superficial. For example, in line 179, what does “as reported by extensive literature” mean? Line 241, “This aquaporin seems to act modifying…? This sentence is basically without content because it leaves the reader without the actual result, such as the AQP2 regulates up or down or whatever is meant. The manuscript lacks precision throughout. Wordings like “high level” in line 241, “peculiar localization” in line 247, “dramatic” in line 248 or the sentence in lines 274-276, starting with “In particular,…”exemplify the point. Several times, “interestingly” is used. Science in general is interesting. Please delete. Such wording does not mean anything. The text requires extensive revision and introduction of not only qualitative but also of quantitative statements. Example, should also indicate the species.
- We are gratefully to the Reviewer for her/his punctual attention in the revision process of the manuscript. As suggested by her/him, we edited several sentences (lines 237-241, lines 322-326) by adding new aspects regarding the cited studies, thus trying to be more detailed about the processes in which AQPs are involved in MSCs functioning. Moreover, we sought to be more accurate in the description of the points above mentioned.
As previous reported, the manuscript has been revised for English grammar by Elsevier English translate service (see Certificate attached).
- In the conclusion section, the authors link AQP2 and stem cells to regenerative medicine but have not addressed regenerative medicine in the manuscript. Therefore, the conclusion from line 281 is not justified.
- We agree with the Reviewer’s comment, about the application of MSCs in the regenerative medicine. Therefore, in the revised version of the manuscript we addressed this topic in the paragraph of MSCs (lines 191-200, lines 225-228, lines 246-250).
Reviewer 2 Report
General Comment: This is an interesting review of the role of AQPs in mesenchymal stem cells.
Major Comment: Role of Aquaporins in the Physiological Functions of Mesenchymal Stem Cells" provided a review of aquaporins in general and of mesenchymal stem cells. The authors provide a balanced review of these topics. The portion of the review discussing the role of aquaporins to physiological functions of mesenchymal stem cells was primarily a review of where aquaporins were located in mesenchymal stem cells. There was little discussion of the physiological role of aquaporins in these cells. If functional data exist in the literature, it should be added to the review. For example, are there any data from aquaporin knockout mice that are pertinent to the physiological function of aquaporins in mesenchymal stem cells. If functional data do not exist, the authors should acknowledge that the literature is in need of functional studies and restate the title of the review to something like "Localization of Aquaporins in Mesenchymal Stem Cells
Minor Comments:
1. The English needs to be carefully reviewed.
2. The are a number of one sentence paragraphs - a paragraph should have a minimum of two sentences
3. line 20 - needs to be rewritten
Author Response
Comments by the reviewer#2
- Role of Aquaporins in the Physiological Functions of Mesenchymal Stem Cells" provided a review of aquaporins in general and of mesenchymal stem cells. The authors provide a balanced review of these topics. The portion of the review discussing the role of aquaporins to physiological functions of mesenchymal stem cells was primarily a review of where aquaporins were located in mesenchymal stem cells. There was little discussion of the physiological role of aquaporins in these cells. If functional data exist in the literature, it should be added to the review. For example, are there any data from aquaporin knockout mice that are pertinent to the physiological function of aquaporins in mesenchymal stem cells. If functional data do not exist, the authors should acknowledge that the literature is in need of functional studies and restate the title of the review to something like "Localization of Aquaporins in Mesenchymal Stem Cells
- We thank the Reviewer’s comment, and we totally agree with her/his consideration. In the revised version of the manuscript, we addressed this issue, trying to better focalize the attention on the physiological role of different AQPs in the modulation of the MSCs functioning. In particular, we added results from studies on knockout mice relative to AQP4 (lines 311-317) and AQP5 (lines 329-332).
- The English needs to be carefully reviewed
- We thank the Reviewer, and the manuscript has been completely revised for English grammar and style by Elsevier English translate service (see certificate attached).
- The are a number of one sentence paragraphs - a paragraph should have a minimum of two sentences
- We perfectly agree with the Reviewer observation and we provided to modify the sentences.
- line 20 - needs to be rewritten
- According to the Reviewer’s suggestion, we edited the sentence at line 20.
Round 2
Reviewer 1 Report
Dear authors
thank you for addressing my concerns.
Reviewer 2 Report
The review is nicely written. It is significantly improved by the additional review of the physiological role of different AQPs in the modulation of the MSC function.